# The cAMP signaling pathway regulates Epe1 protein levels and heterochromatin assembly

Kehan Bao[1], Chun-Min Shan[1¤], Xiao Chen[2,3], Gulzhan Raiymbek[4], Jeremy G. Monroe[5], Yimeng Fang[1], Takenori Toda[1], Kristin S. Koutmou[5], Kaushik Ragunathan[4], Chao Lu[2,3], Luke E. Berchowitz[2], Songtao Jia[1]*

1 Department of Biological Sciences, Columbia University, New York, New York, United States of America, 2 Department of Genetics and Development, Columbia University Irving Medical Center, New York, New York, United States of America, 3 Herbert Irving Comprehensive Cancer Center, Columbia University Irving Medical Center, New York, New York, United States of America, 4 Department of Biological Chemistry, University of Michigan, Ann Arbor, Michigan, United States of America, 5 Department of Chemistry, University of Michigan, Ann Arbor, Michigan, United States of America

¤ Current address: State Key Laboratory of Plant Genomics, Institute of Microbiology, Chinese Academy of Sciences, Beijing, China
* songtao.jia@columbia.edu

**Data Availability Statement:** ChIP-sequencing data that support the findings of this study have been deposited in NCBI GEO with the accession number: GSE181263.

## Abstract

The epigenetic landscape of a cell frequently changes in response to fluctuations in nutrient levels, but the mechanistic link is not well understood. In fission yeast, the JmjC domain protein Epe1 is critical for maintaining the heterochromatin landscape. While loss of Epe1 results in heterochromatin expansion, overexpression of Epe1 leads to defective heterochromatin. Through a genetic screen, we found that mutations in genes of the cAMP signaling pathway suppress the heterochromatin defects associated with Epe1 overexpression. We further demonstrated that the activation of Pka1, the downstream effector of cAMP signaling, is required for the efficient translation of $epe1^+$ mRNA to maintain Epe1 overexpression. Moreover, inactivation of the cAMP-signaling pathway, either through genetic mutations or glucose deprivation, leads to the reduction of endogenous Epe1 and corresponding heterochromatin changes. These results reveal the mechanism by which the cAMP signaling pathway regulates heterochromatin landscape in fission yeast.

## Author summary

Genomic DNA is folded with histones into chromatin and posttranslational modifications on histones separate chromatin into active euchromatin and repressive heterochromatin. These chromatin domains often change in response to environmental cues, such as nutrient levels. How environmental changes affect histone modifications is not well understood. Here, we found that in fission yeast, the cAMP signaling pathway is required for the function of Epe1, an enzyme that removes histone modifications associated with heterochromatin. Moreover, we found that active cAMP signaling ensures the efficient translation of $epe1^+$ mRNA and therefore maintains high Epe1 protein levels. Finally, we show that changing glucose levels, which modulate cAMP signaling, also affect heterochromatin

**Funding:** This work was supported by NIH grants R35GM126910 to S.J., and R35GM138181 to C.L., The Sachaefer Research Program to LEB, The Hirschl Family Trust to LEB, NIH grant R35GM124633 to L.E.B, NIH grant R35GM128836 to KK, and NIH grant R35GM137832 to KR. The funders had no role in study design, data collection and analysis, decision to publish, or preparation of the manuscript.

**Competing interests:** The authors have declared that no competing interests exist.

in a way consistent with cAMP signaling-mediated Epe1 protein level changes. As histone-modifying enzymes often require cofactors that are metabolic intermediates, previous studies on the impact of nutrient levels on chromatin states have mainly focused on metabolites. Our results suggest that nutrient-sensing signaling pathways also regulate histone-modifying enzymes in response to nutritional conditions.

## Introduction

Genomic DNA is folded with histones and non-histone proteins into chromatin, and post-translational modifications of histones play major roles in regulating genome function. Based on histone modification profiles and other characteristics, chromatin is classified into two main categories: euchromatin and heterochromatin. Euchromatin is gene-rich, transcriptionally active, less condensed, and enriched with histones that are hyperacetylated. On the other hand, heterochromatin is gene-poor, transcriptionally silent, more compact, and enriched with histones that are hypoacetylated and methylated at H3 lysine 9 (H3K9me) or H3 lysine K27 (H3K27me) [1,2]. These chromatin states are relatively stable, but they also change dynamically in response to environmental stimuli [3,4]. However, the signaling events that transduce outside signals to chromatin are not well understood.

In the fission yeast *Schizosaccharomyces pombe*, large blocks of heterochromatin form at repetitive DNA elements near centromeres, telomeres, and at the silent mating-type region. Additionally, about two dozen small heterochromatin islands are scattered throughout the genome [1,2]. While large heterochromatin domains are relatively stable, heterochromatin islands frequently change in response to diverse environmental conditions. For example, upon nitrogen starvation, heterochromatin islands are dissembled at meiotic genes as cells prepare for sexual differentiation [5]. In addition, at low temperatures, heterochromatin islands change dramatically in an iron-dependent manner to fine-tune the transcription response [6]. Interestingly, challenges caused by certain genetic mutations or drugs also allow the formation of ectopic heterochromatin islands to create epigenetically silenced gene alleles that enable cells to survive [7,8]. How heterochromatin changes in response to environmental conditions is not clear, but the myriad of proteins that participate in the formation and erasure of heterochromatin provide ample targets for signaling pathways to relay environmental information.

The formation of heterochromatin in fission yeast depends on diverse pathways that recruit the histone H3K9 methyltransferase Clr4 to distinct genomic locations [1,2]. For example, repetitive DNA elements recruit Clr4 through the RITS (RNA induced transcriptional silencing) complex, which uses small interfering RNAs (siRNAs) generated by the RNA interference (RNAi) machinery as guides to home in on the nascent transcripts produced at repeat regions [9–13]. In addition, DNA binding proteins, such as ATF/CREB proteins Atf1/Pcr1 and shelterin, recruit Clr4 to nucleate heterochromatin near repeats at the silent mating-type region and telomeres, respectively [14–16]. Moreover, the RNA elimination machinery recognizes nascent RNAs containing determinant of selective removal (DSR) sequence to recruit Clr4 and establish heterochromatin islands [5]. Clr4-mediated H3K9 methylation leads to the recruitment of HP1 proteins to create a silenced chromatin state that represses both transcription and recombination [1,17–19].

Maintaining the proper heterochromatin landscape also depends on diverse activities that remove heterochromatin from inappropriate sites, such as the Mst2 histone H3K14 acetyltransferase complex, the INO80 chromatin remodeling complex, and the JmjC domain protein Epe1 [5,7,20]. Epe1 is a resident heterochromatin protein and is recruited to heterochromatin

through its interaction with Swi6 [21–23]. It contains a JmjC domain, which is the catalytic domain of histone demethylases [24]. Although no demethylase activity has been demonstrated for Epe1 in vitro [24], in vivo evidence demonstrates that Epe1 is the major "eraser" of H3K9me. For example, loss of Epe1 leads to spreading of heterochromatin outside of its normal boundaries, expansion of heterochromatin islands, formation of ectopic heterochromatin, and more stable inheritance of heterochromatin. In contrast, overexpression of Epe1 leads to the destabilization of existing heterochromatin [5,7,21,22,25–29]. Therefore, Epe1 protein levels need to be tightly regulated within a narrow range. Indeed, Epe1 is a target of the Cul4-Ddb1 ubiquitin E3 ligase, which mediates its degradation by the proteasome [30]. Epe1 is required for heterochromatin island changes in response to nitrogen starvation and loss of Epe1 function has been implicated in the generation of new epigenetically silenced alleles [5,7,8,29]. However, how Epe1 links changes in the heterochromatin landscape to environmental conditions is still unknown.

In this study, we performed a genetic screen to identify regulators of Epe1 function. We found that an active cAMP signaling pathway is critical for the ability of Epe1 to erase heterochromatin. We further demonstrated that the cAMP signaling pathway maintains Epe1 protein levels by regulating its mRNA translation. These results provide a critical link between nutritional conditions and the heterochromatin landscape of the genome.

## Results

### The cAMP signaling pathway regulates Epe1 function

Given the critical role of Epe1 in modulating the heterochromatin landscape, we performed a genetic screen for regulators of Epe1 function. We replaced the endogenous promoter of *epe1*$^+$ with an *nmt41* promoter, which can be induced to overexpress Epe1 when cells are grown in a medium without thiamine (Edinburgh minimal medium, EMM) [31]. We also used a reporter gene inserted within pericentric repeats (*otr:ura4*$^+$) to measure heterochromatin integrity [32]. In wild-type cells, *otr:ura4*$^+$ is silenced by the formation of heterochromatin, resulting in cells resistant to 5-fluoroorotic acid (5-FOA). When Epe1 is overexpressed, heterochromatin is compromised, leading to the expression of the *otr:ura4*$^+$ reporter and thus poor cell growth on media containing 5-FOA [21,22,28].

We crossed a query strain containing both the *otr::ura4*$^+$ reporter and *nmt41-epe1*$^+$ with the fission yeast deletion library, which contains ~3500 nonessential gene deletions (Fig 1A). After selecting haploid progeny containing *otr::ura4*$^+$, *nmt41-epe1*$^+$, and the deletion of a single gene, cells were grown on EMM medium containing 5-FOA to measure the silencing of *otr:: ura4*$^+$. We found that deletions of several genes involved in the cAMP (cyclic adenosine 3',5'-monophosphate)-signaling pathway, such as *git1Δ*, *git3Δ*, *git5Δ*, *gpa2Δ*, and *pka1Δ*, rescue the silencing defect caused by Epe1 overexpression (Fig 1B).

The cAMP signaling pathway coordinates cellular responses with outside stimuli, such as hormones and nutrients, and is largely conserved from yeast to mammals [33,34] (Fig 1C). In fission yeast, ligand molecules bind to the transmembrane G protein-coupled receptor (GPCR) Git3, which in turn activates the G protein trimers Gpa2/Git5/Git11, releasing the G-alpha subunit Gpa2 to activate the adenylyl cyclase Cyr1 to synthesize cAMP. Protein kinase A (Pka1) is the major effector of the cAMP signaling pathway in fission yeast. In the absence of cAMP, Pka1 is inactive due to its association with a regulatory subunit, Cgs1. In the presence of cAMP, Cgs1 dissociates from Pka1. Pka1 then translocates to the nucleus and phosphorylates its substrates. The identification of multiple mutants of the cAMP signaling pathway in our screen indicates that this pathway regulates Epe1 function.

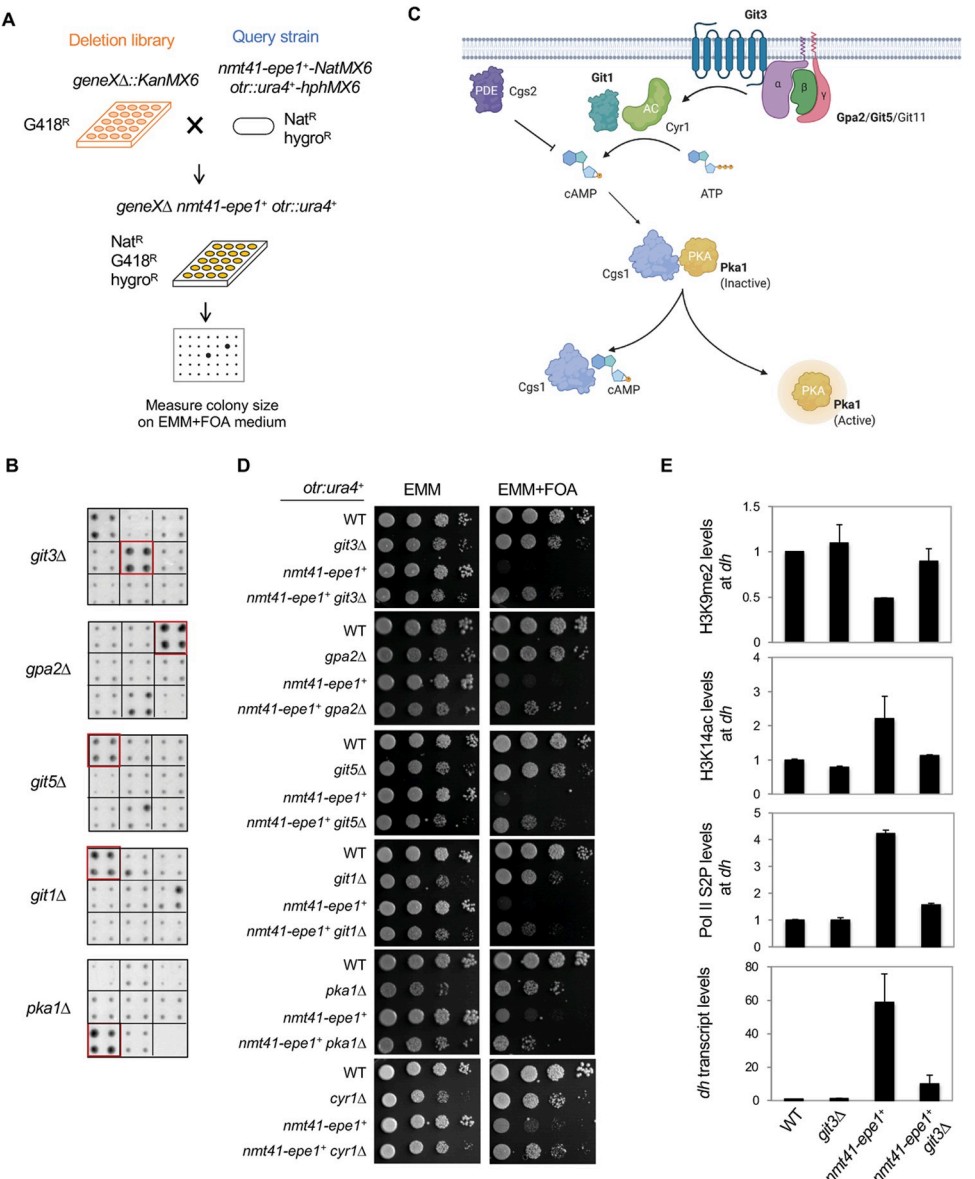

**Fig 1. A genetic screen for regulators of Epe1 function.** (A) Workflow to introduce *otr::ura4+* and *nmt41-epe1+* into the deletion library. (B) Representative images of cells grown on EMM+FOA medium. Each square represents one gene deletion in quadruplicates. Red boxes represent cells with genotypes marked on the left. (C) Diagram of the cAMP signaling pathway in *S. pombe*. The factors identified in our screen are in bold. (D) Ten-fold serial dilution analyses of indicated yeast strains grown on indicated media to measure the expression of the *otr::ura4+* reporter. (E) ChIP analyses of H3K9me2, H3K14ac, and Pol II S2P levels at pericentric repeats (*dh*), normalized to *leu1+*, and RT-qPCR analysis of pericentric transcript (*dh*), normalized to *act1+*. Error bars represent SD, n≥3.

To validate the results of our genetic screen, we reconstructed cells containing *otr::ura4+ nmt41-epe1+* and individual gene deletions by genetic crosses. We also constructed *cyr1Δ*, which is not present in the deletion library. Consistent with the results of the genetic screen, serial dilution analyses show that *git1Δ*, *git3Δ*, *git5Δ*, *gpa2Δ*, *pka1Δ*, and *cyr1Δ* all rescue silencing defects of *otr::ura4+* caused by Epe1 overexpression, as indicated by better growth on EMM medium containing 5-FOA (Fig 1D).

We then examined the effects of cAMP signaling on heterochromatin at pericentric repeats, which are divided into *dh* and *dg* regions. Pericentric heterochromatin is enriched in repressive histone posttranslational modifications such as H3K9 methylation. Epe1-overexpression not only results in the loss of silencing of reporter genes inserted within pericentric heterochromatin but also a reduction of H3K9 methylation levels at *dh* repeats [21,22,28]. ChIP analysis shows that H3K9me2 levels at *dh* repeats are restored close to wild-type levels in *git3Δ nmt41-epe1*+ cells (Fig 1E).

Epe1 overexpression recruits the SAGA histone acetyltransferase to pericentric repeats, leading to increases in H3K14ac and Ser2-phosphorylated Pol II levels at heterochromatin, as well as increases in *dh* transcripts levels [28]. ChIP analysis reveals that both H3K14ac and Ser2-phosphorylated Pol II are restored to near wild-type levels in *git3Δ nmt41-epe1*+ cells (Fig 1E). Moreover, RT-qPCR analysis shows that the pericentric repeat transcript levels are elevated in *nmt41-epe1*+ cells, but are reduced to near wild-type levels in *git3Δ nmt41-epe1*+ cells (Fig 1E). These results support a model whereby active cAMP signaling mediates the heterochromatin defects caused by Epe1 overexpression.

## Intracellular cAMP levels regulate Epe1 function

We next examined whether intracellular cAMP levels regulate Epe1 function. In fission yeast, the transmembrane G-protein-coupled receptor Git3 and the downstream heterotrimeric G proteins (Gpa2, Git5, and Git11) activate the adenylate cyclase Cyr1 to raise intracellular cAMP levels [33]. Cgs2 is a phosphodiesterase that breaks down cAMP, and *cgs2Δ* raises intracellular cAMP levels in the absence of Git3 [35] (Fig 2A). Interestingly, although *git3Δ nmt41-epe1*+ cells form heterochromatin at pericentric repeats, *cgs2Δ git3Δ nmt41-epe1*+ cells do not, as indicated by both serial dilution analysis to measure the expression of *otr::ura4*+ and RT-qPCR analysis to measure *dh* transcript levels (Fig 2B and 2C). These results suggest that the reduction of cellular cAMP levels caused by *git3Δ* is responsible for the rescue of the Epe1 overexpression phenotype.

To further examine the role of cAMP in this process, we directly added cAMP to the growth medium. RT-qPCR analysis shows that *dh* transcript levels of *git3Δ nmt41-epe1*+ cells are almost at wild-type levels when grown in EMM medium, but they are higher in the presence of 5mM exogenous cAMP (Fig 2D), consistent with the notion that cAMP levels regulate Epe1 function.

## Activation of Pka1 is required for Epe1 function

We then examined whether cAMP affects Epe1 function via the effector kinase Pka1. When cAMP levels are low, Pka1 is inactive due to the association of its regulatory subunit Cgs1. In *cgs1Δ* cells, Pka1 is constitutively active regardless of intracellular cAMP levels [36] (Fig 2A). Similar to *cgs2Δ*, *cgs1Δ* also reverts the rescue of heterochromatin in *git3Δ nmt41-epe1*+ cells, as measured by serial dilution analysis to measure *otr::ura4*+ expression and RT-qPCR analysis to measure *dh* transcript levels (Fig 2B and 2C). In contrast, *cgs2Δ*, *cgs1Δ*, or the addition of cAMP have mild effects on *dh* transcript levels in *pka1Δ nmt41-epe1*+ cells (Fig 2E). These results support the idea that cAMP activates Pka1, which in turn stimulates Epe1 function.

## cAMP signaling pathway regulates Epe1 protein levels

To further examine how the cAMP signaling pathway regulates Epe1 function, we assessed the effects of cAMP signaling mutations on Epe1 protein levels. Interestingly, we found that Epe1 protein levels are significantly reduced in *git3Δ nmt41-epe1*+ and *pka1Δ nmt41-epe1*+ cells (Fig 2F). In addition, Epe1 protein levels are largely restored to wild-type levels in *git3Δ cgs1Δ*

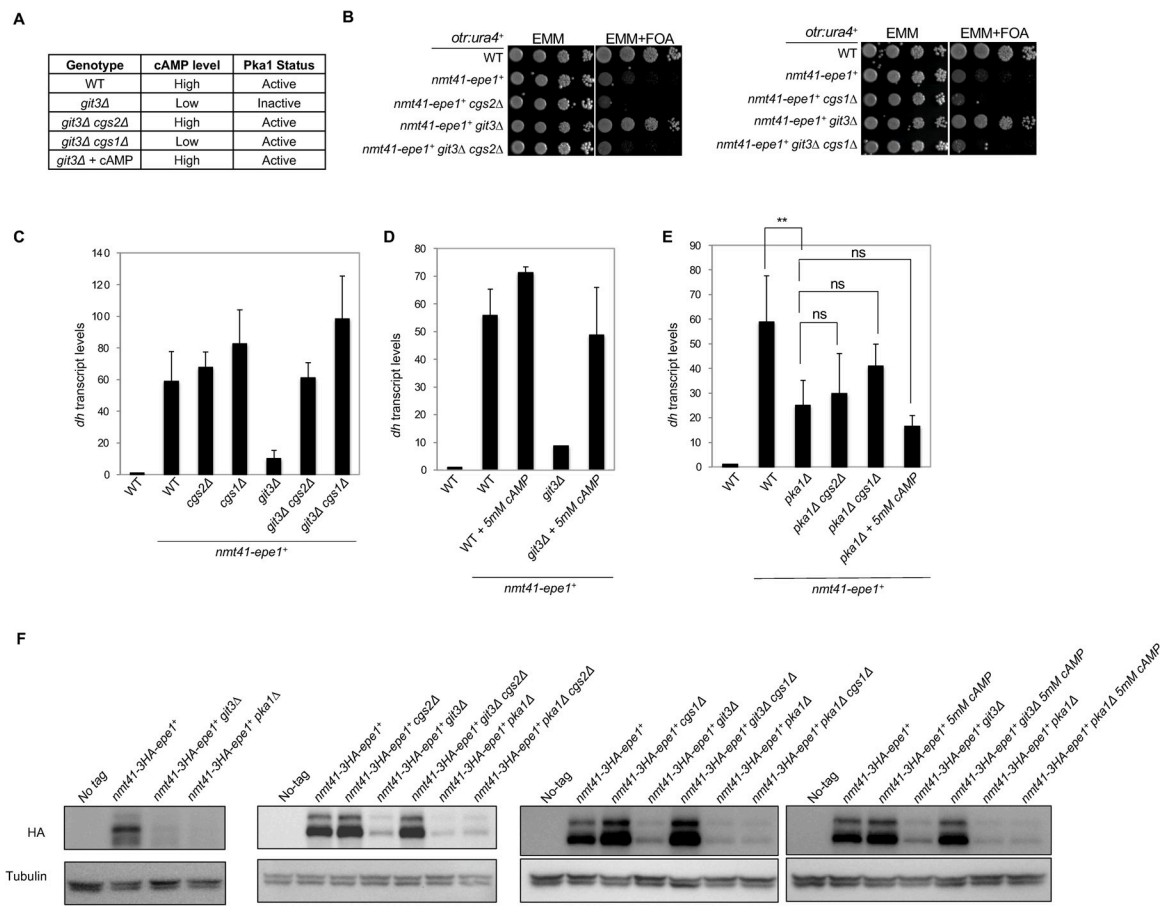

**Fig 2. High levels of cAMP and an active Pka1 are required for Epe1 function.** (A) Summary of effects of *git3Δ*, *cgs2Δ*, *cgs1Δ*, and exogenous cAMP on cellular cAMP levels and the activity of Pka1. (B) Ten-fold serial dilution analyses of indicated yeast strains grown on indicated media to measure the expression of the *otr::ura4⁺* reporter gene. (C-E) RT-qPCR analyses of pericentric *dh* transcript, normalized to *act1⁺*. Error bars represent SD, n≥2. One-way ANOVA was performed comparing each strain to *nmt41-epe1⁺ pka1Δ*. (F) Western blot analyses to measure the levels of 3HA-tagged Epe1 under the control of *nmt41* promoter and the levels of tubulin.

*nmt41-epe1⁺* and *git3Δ cgs2Δ nmt41-epe1⁺* cells (Fig 2F). Moreover, 5mM exogenous cAMP also restores Epe1 protein levels in *git3Δ nmt41-epe1⁺* cells (Fig 2F). On the other hand, *cgs1Δ*, *cgs2Δ*, and 5mM exogenous cAMP do not restore Epe1 levels in *pka1Δ* cells (Fig 2F). These data suggest that the cAMP signaling pathway governs Epe1 protein homeostasis.

## Epe1 phosphorylation by Pka1 does not contribute to Epe1 protein level control

One possible explanation of how cAMP signaling regulates Epe1 protein levels is that Epe1 is a direct target of Pka1 phosphorylation. Indeed, we found that Pka1 phosphorylates recombinant Epe1 *in vitro*, although the activity is much weaker compared to another Pka1 substrate Rst2 (S1A and S1B Fig). We subjected *in vitro* phosphorylated Epe1 to mass-spec analysis and identified S717 as the only phosphorylation site. We mutated S717 to a phosphomimetic amino acid (S717D), but this mutation did not protect Epe1 levels from decreasing in *git3Δ* cells (S1C Fig). In addition, bioinformatics analysis also predicted Pka1 phosphorylation sites at residues S606 and T607. We mutated these two residues to phosphomimetic amino acids

(S606D T607D), but the protein levels of this Epe1 mutant also decrease in *git3Δ* cells (S1D Fig). These results suggest that Pka1-mediated phosphorylation of Epe1 might not be responsible for Epe1 protein level changes in the absence of active cAMP signaling.

## cAMP signaling pathway regulates Epe1 protein levels through translation control

The cAMP signaling pathway may also regulate Epe1 protein levels through transcription, translation, or protein degradation. To distinguish these possibilities, we first measured *epe1+* transcript levels by RT-qPCR. However, we did not observe any reduction of *epe1+* mRNA levels in *git3Δ* or *pka1Δ* cells (Fig 3A), indicating that cAMP signaling regulates Epe1 protein levels through a post-transcriptional mechanism.

Because Epe1 protein levels are regulated by the Cul4-Ddb1 E3 ubiquitin ligase complex [30], we tested whether cAMP signaling regulates Cul4-Ddb1-mediated Epe1 degradation. There are severe growth defects associated with *ddb1Δ* due to the accumulation of Spd1, one of Ddb1's targets [37]. Therefore, we used *ddb1Δ spd1Δ* cells to avoid complications from slow growth. Epe1 protein levels remain low in *git3Δ ddb1Δ spd1Δ* cells, similar to those in *git3Δ* cells, suggesting that the cAMP signaling pathway does not regulate Epe1 degradation through Cul4-Ddb1 (Fig 3B).

To further examine whether cAMP signaling regulates Epe1 degradation, we measured Epe1 degradation kinetics in wild-type and *git3Δ* cells after the addition of cycloheximide

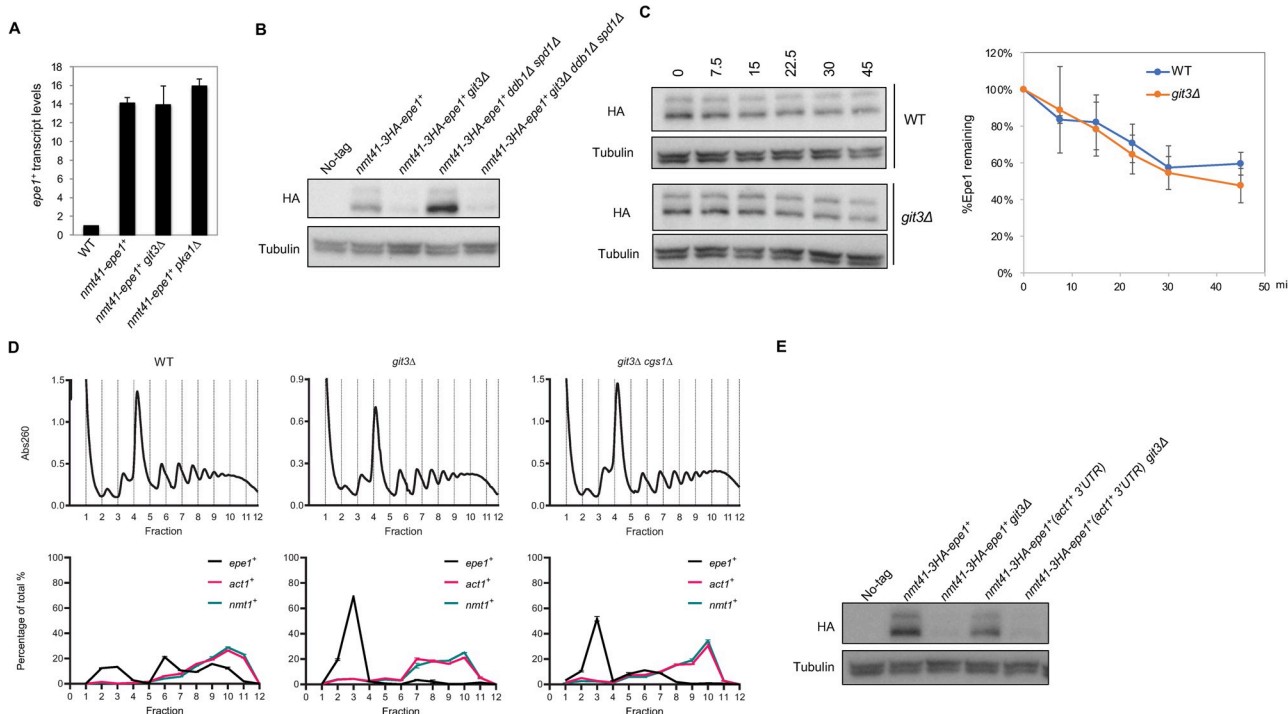

**Fig 3. The cAMP signaling pathway regulates the translation of *epe1+* mRNA.** (A) RT-qPCR analysis of *epe1+* transcript, normalized to *act1+*. Error bars represent SD, n = 2. (B, E) Western blot analysis to measure the levels of 3HA-tagged Epe1 under the control of *nmt41* promoter and levels of tubulin. (C) Left, western blot analysis to measure the levels of 3HA-tagged Epe1 under the control of *nmt41* promoter and tubulin. Samples are taken at indicated time points (minutes) after the addition of cycloheximide. Different exposures were used for wild type and *git3Δ* cells to make the signal at time 0 comparable. Right, quantification of Epe1 proteins levels. Error bars represent SD, n = 2. (D) Top, sucrose density gradients of ribosomes in each strain with continuous monitoring of absorbance at 260 nm. Lighter fractions are on the left. Bottom, qRT-PCR analysis of *epe1+*, *act1+*, and *nmt1+* transcripts from each fraction. Relative amounts of transcripts were calculated using the delta Ct method. The distribution is shown as a percentage of the total. Error bars represent the SD of two technical replicates.

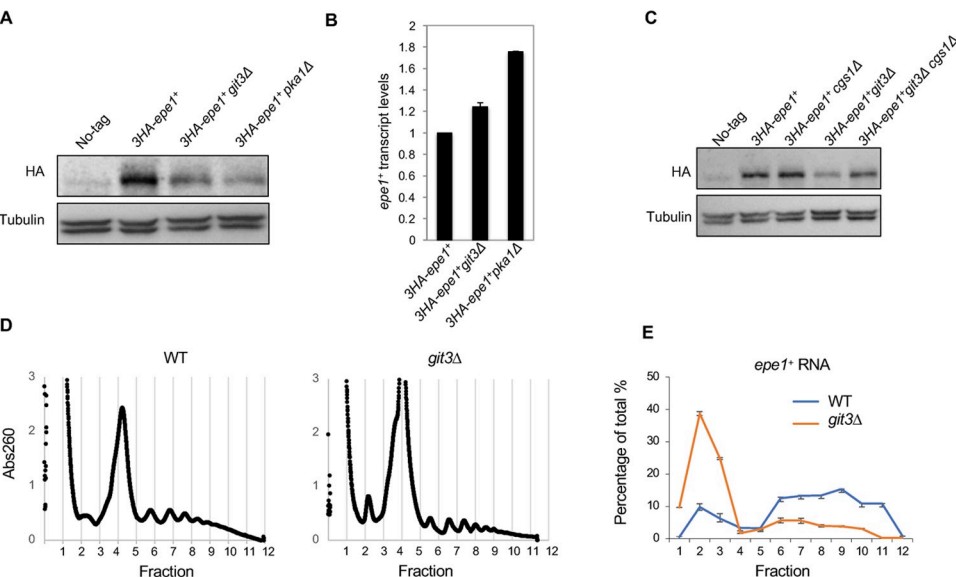

**Fig 4. The cAMP signaling pathway regulates the protein levels of endogenous Epe1.** (A, C) Western blot analysis to measure the levels of endogenous 3HA-tagged Epe1 and tubulin. (B) RT-qPCR analysis of $epe1^+$ transcript levels, normalized to $act1^+$. Error bars represent SD. n = 2. (C) Sucrose density gradients of ribosomes in each strain with continuous monitoring of absorbance at 260 nm. Lighter fractions are on the left. (E) qRT-PCR analysis of $epe1^+$ transcripts from each fraction. Relative amounts of transcripts were calculated using the delta Ct method. The distribution is shown as a percentage of the total. Error bars represent the SD of two technical replicates.

(CHX) to block new protein synthesis. The degradation rates of Epe1 are similar over a 45-minute period after CHX addition, indicating that the cAMP signaling pathway does not control Epe1 levels by regulating its degradation (Fig 3C).

We then examined whether cAMP signaling regulates the translation of Epe1 using polysome profiling. In $nmt41$-$epe1^+$ cells grown in EMM medium, $epe1^+$ mRNA is broadly distributed into different fractions. In contrast, an actively transcribed housekeeping gene $act1^+$, which encodes actin, is mainly present in the polysome fractions (5 through 12) (Fig 3D). Remarkably, $git3\Delta$ nearly abolished $epe1^+$ mRNA in the polysome fractions, and $git3\Delta$ $cgs1\Delta$ partially restores polysome-associated $epe1^+$ mRNA (Fig 3D). These results suggest that cAMP signaling regulates $epe1^+$ mRNA translation.

We then assessed whether and to what degree the cAMP-mediated effects on Epe1 translation were dependent on the untranslated regions (UTRs). The $nmt41$-$epe1^+$ construct replaces the endogenous $epe1^+$ promoter with a $nmt41$ promoter, which contains a 4 base pair deletion at the TATA box of the endogenous $nmt1$ promoter to reduce its expression [31,38]. The 5'-UTR of $nmt41$-$epe1^+$ is identical to that of $nmt1^+$. The endogenous $nmt1^+$ mRNA, which is also induced to express at high levels, shows a distribution in polysome profile similar to that of $act1^+$ mRNA, and the distribution is not severely affected by either $git3\Delta$ or $git3\Delta$ $cgs1\Delta$ (Fig 3D). We also replaced the 3'-UTR of $nmt41$-$epe1^+$ with the 3'-UTR of $act1^+$ and found that Epe1 protein levels are still reduced in $git3\Delta$ cells (Fig 3E). These results strongly argue against a role for the 5'-UTR or 3'-UTR regions in regulating $epe1^+$ mRNA translation.

## The cAMP signaling pathway regulates endogenous Epe1 protein levels and heterochromatin

We then tested if the cAMP signaling pathway regulates Epe1 expressed from the endogenous locus. We inserted three copies of HA tag at the N-terminus of the $epe1^+$ at its endogenous

chromosomal locus, keeping the promoter, 5'UTR, and 3'-UTR intact. We first tested whether the addition of the HA tag affects Epe1 function using an *ade6*⁺ reporter inserted outside of mating-type region heterochromatin (*SacI::ade6*⁺) (S2A Fig). This reporter is fully expressed in wild-type cells, resulting in white colonies when cells are grown on a low adenine medium (YE) [25] (S2B Fig). In *epe1Δ* cells, heterochromatin spreads outside of the boundary to silence *SacI::ade6*⁺, resulting in red colonies. The addition of the HA-tag does not significantly compromise Epe1 function, as HA-Epe1 expressing cells form mostly white colonies (S2B Fig). In both *git3Δ* and *pka1Δ* cells, HA-Epe1 protein levels are reduced without a corresponding decrease in *epe1*⁺ mRNA levels (Fig 4A and 4B). Moreover, Epe1 protein levels increase in *git3Δ cgs1Δ* cells compared with *git3Δ* cells (Fig 4C). These results suggest that endogenous Epe1 is regulated by cAMP signaling. Finally, polysome profiling shows that endogenous *epe1*⁺ mRNA is broadly distributed into different fractions in wild-type cells. However, its presence in the polysome fractions is reduced in *git3Δ* cells (Fig 4D and 4E). These results suggest that cAMP signaling also specifically regulates endogenous *epe1*⁺ mRNA translation, irrespective of its expression levels. In contrast, there are minor differences in the distribution of *clr4*⁺ mRNA in polysome fractions between wild type and *git3Δ* cells (S3A Fig), and Clr4 protein levels are not affected by *git3Δ* (S3B Fig)

## The cAMP signaling pathway regulates Epe1 function in heterochromatin assembly

Since Epe1 levels are reduced when the cAMP signaling pathway is inactive, we examined if *git3Δ* phenocopies *epe1Δ* in heterochromatin regulation. Epe1 was originally identified as a factor required for confining the silent mating-type region heterochromatin within proper boundaries and *epe1Δ* results in the silencing of the *SacI::ade6*⁺ reporter [25]. In *git3Δ* cells, *SacI::ade6*⁺ is also silenced, leading to red/pink colonies, although the effect is weaker compared with *epe1Δ* (Fig 5A). RT-qPCR analyses of the *ade6*⁺ transcript also show a stronger silencing effect in *epe1Δ* cells than in *git3Δ* cells, and *epe1Δ git3Δ* cells behave similarly to *epe1Δ* cells (Fig 5A). In addition, serial dilution and RT-qPCR analyses show that *git3Δ cgs1Δ* reduced heterochromatin spreading at *SacI::ade6*⁺ compared with *git3Δ* (Fig 5A), consistent with Epe1 protein level changes in these cells (Fig 4C).

The fission yeast genome contains a number of small heterochromatin islands that exhibit varying levels of H3K9me2 and in *epe1Δ* cells, H3K9me2 levels increase at a majority of these islands [5]. ChIP analyses show that H3K9me2 levels increase to a similar extent at two major heterochromatin islands, *mei4*⁺ and *ssm4*⁺, in *git3Δ*, *epe1Δ*, and *git3Δ epe1Δ* cells (Fig 5B).

Epe1 also regulates heterochromatin inheritance [39,40]. During DNA replication, the passage of the replication fork disrupts parental nucleosomes. Parental (H3-H4)₂ tetramers, which are marked by H3K9me3, are deposited at the original location and to both daughter strands to direct the formation of nucleosomes. The remaining gaps in DNA are filled by nucleosomes formed with newly synthesized (H3-H4)₂. The H3K9m3 on parental histones recruits Clr4, which contains a chromodomain that recognizes H3K9me3. Clr4 then methylates nearby nucleosomes containing newly synthesized histones, therefore restoring the original histone modification profiles on both replicated DNA strands [41]. Since most native heterochromatin regions contain signals for the recruitment of Clr4, ectopic heterochromatin systems have been developed to specifically examine heterochromatin inheritance in the absence of initiation signals [39,40]. For example, when the SET domain of Clr4 is targeted to ten copies of *tetO* binding sites through a TetR fusion protein (TetR-Clr4-SET), the formation of a large heterochromatin domain silences a neighboring GFP reporter gene (*tetO-gfp*⁺) [40] (S4A Fig). The addition of tetracycline to the medium leads to the quick release of

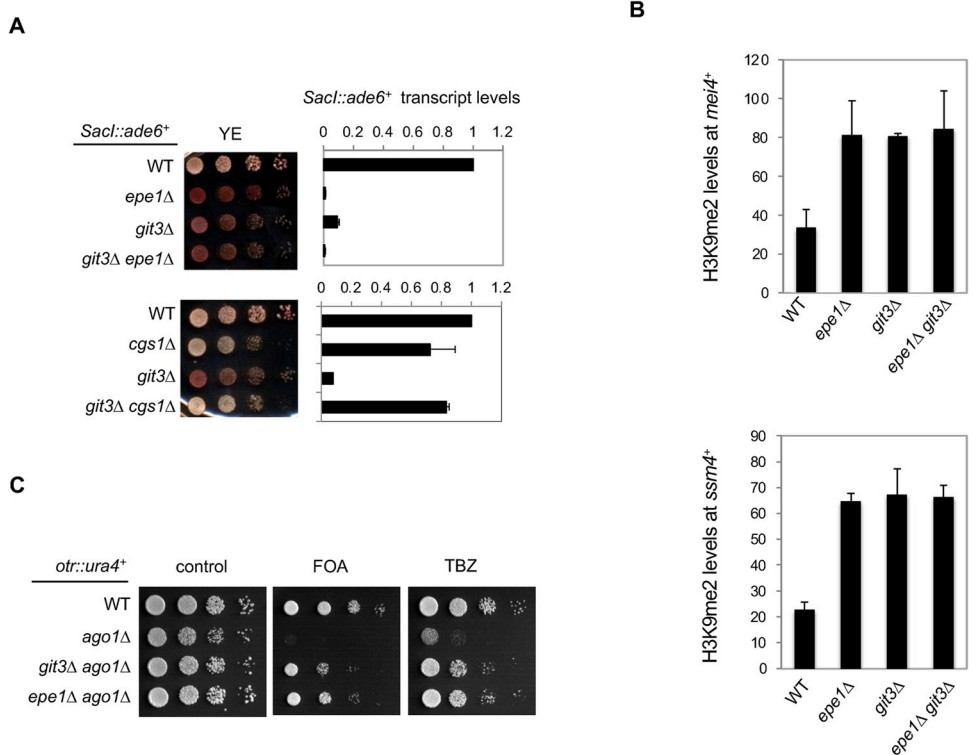

**Fig 5. The cAMP signaling pathway regulates heterochromatin formation.** (A) Left, Ten-fold serial dilution analyses to measure the expression of the *SacI::ade6+* reporter gene. Right, RT-qPCR analysis of *ade6+* transcript, normalized to *act1+*. The endogenous *ade6+* in these strains contains an internal deletion (DN/N). Primers were designed to amplify the reporter *ade6+* but not the endogenous *ade6-DN/N*. Error bars represent SD, n = 2. (B) ChIP analyses of H3K9me2 levels at two prominent heterochromatin islands. Error bars represent SD, n = 3. (C) Ten-fold serial dilution analyses to measure the expression of *otr::ura4+* and sensitivity to TBZ.

TetR-Clr4-SET. Endogenous Clr4 is recruited to regions with preexisting H3K9me3 and methylates newly incorporated histones due to replication-coupled nucleosome assembly or histone turnover, resulting in the inheritance of this ectopic heterochromatin (S4A Fig). In wild-type cells, this inheritance mechanism is hindered by Epe1-mediated erasure of H3K9me3. As a result, heterochromatin decays quickly, and fluorescence-activated cell sorting (FACS) shows that the expression of GFP gradually increases over a 24-hour period after tetracycline addition (S4B Fig). In *epe1Δ* cells, the majority of cells still silence GFP expression 24 hours after tetracycline addition. However, *git3Δ* results in defective silencing of the *tetO-gfp+* reporter even before the addition of tetracycline (S4B Fig), making it difficult to assess the effects of cAMP signaling in heterochromatin inheritance using this system. The reason that *git3Δ* causes silencing defects at *tetO-gfp+* is unclear. The cAMP signaling pathway may regulate other factors that indirectly affect the silencing at the reporter locus.

To overcome this complication, we examined the effects of the cAMP signaling pathway on the inheritance of pericentric heterochromatin in the absence of RNAi, which is also dependent on the inactivation of Epe1 [41]. For instance, *ago1Δ* results in the loss of silencing of the *otr::ura4+* reporter, and cells are sensitive to microtubule poison thiabendazole (TBZ) due to the requirement of pericentric heterochromatin for chromosome segregation. In *epe1Δ ago1Δ* cells, both silencing of *otr::ura4+* and TBZ sensitivity are restored [22,40] (Fig 5C). Consistent with the idea that cAMP signaling regulates Epe1 function, *git3Δ ago1Δ* cells also partially rescue *otr::ura4+* silencing defects and TBZ sensitivity associated with *ago1Δ* (Fig 5C)

## Low glucose treatment reduces Epe1 protein levels and changes heterochromatin landscape

In fission yeast, the cAMP signaling pathway is active when cells are grown in a rich medium such as YEA (3% glucose) but inactive when glucose is scarce. Epe1 protein levels decrease after 6 hours of growth in a low glucose medium (0.1% glucose, 3% glycerol), even though *epe1*$^+$ mRNA levels increase (Fig 6A and 6B), suggesting that Epe1 protein levels are controlled by a post-transcriptional mechanism in low glucose conditions as well. In addition, Epe1 protein levels are partially restored in *cgs1Δ* cells subjected to low glucose treatment (S5A and S5B Fig).

We then measured the effects of low glucose medium on heterochromatin functions. qRT-PCR analyses show that 6 hours of low glucose treatment results in about 50% reduction of *SacI::ade6*$^+$ expression in wild-type cells, suggesting increased heterochromatin spreading.

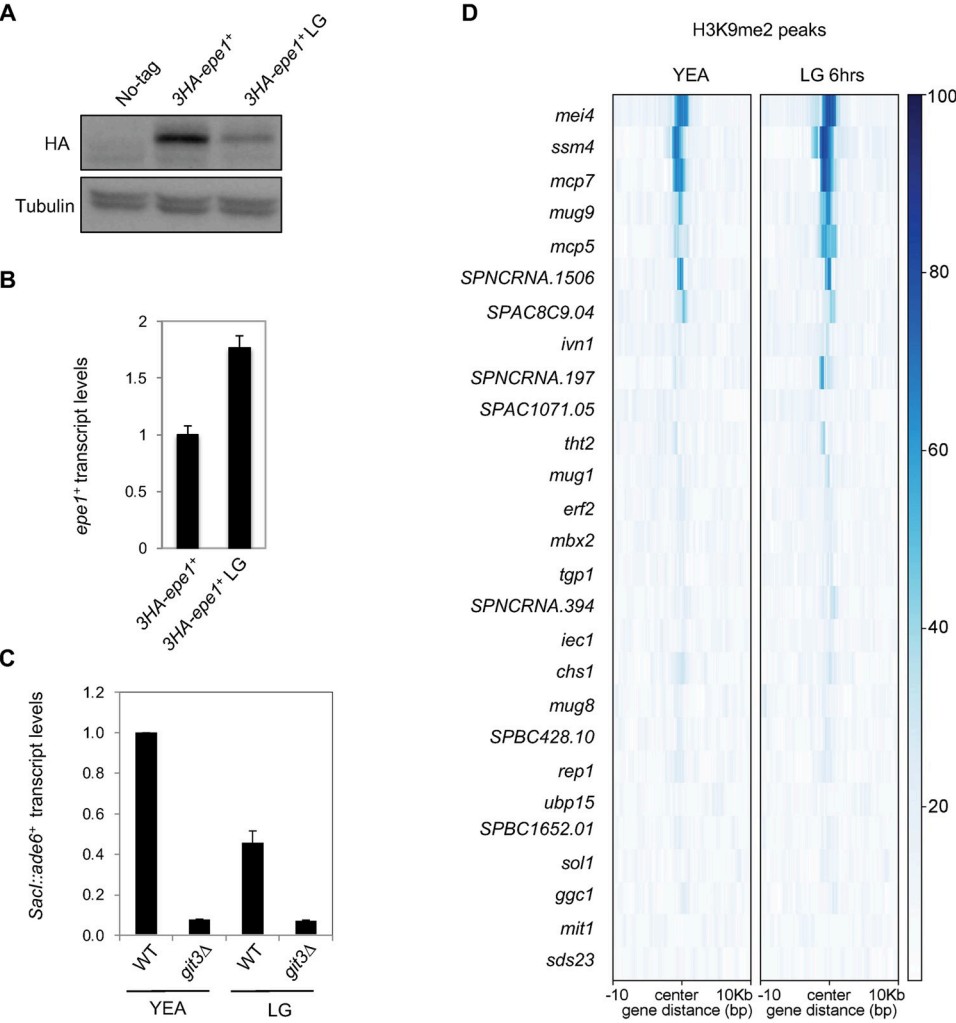

**Fig 6. Extracellular glucose concentration affects Epe1 protein levels and heterochromatin formation.** (A) Western blot analysis to measure the levels of 3HA-tagged Epe1 and tubulin. Cells were shifted to a low-glucose medium (LG) for 6 hours before protein extraction. (B) RT-qPCR analysis of *epe1*$^+$ transcript levels, normalized to *act1*$^+$. Error bars represent SD, n = 2. (C) RT-qPCR analysis of *ade6*$^+$ transcript, normalized to *act1*$^+$. Error bars represent SD, n = 2. (D) Heatmap of H3K9me2 levels at individual heterochromatin islands.

In contrast, such treatment has little impact on *SacI::ade6+* expression in *git3Δ* cells, suggesting that the reduction in wild-type cells is due to decreased cAMP signaling (Fig 6C). In addition, FACS analysis shows that heterochromatin at the *tetO-gfp+* reporter decays slower in cells grown in low glucose medium after tetracycline addition to remove tetR-Clr4-SET (S4C Fig). However, we note that there is decreased silencing of the reporter before tetracycline addition and cells grow slower in low glucose medium, therefore making this result difficult to interpret. Finally, ChIP-seq analyses show that H3K9me2 levels at many heterochromatin islands appear to increase in cells grown in low glucose medium for 6 hours than those grown in high glucose medium (YEA) (Fig 6D). These results together suggest that extracellular glucose concentration may regulate heterochromatin functions through Epe1.

## Discussion

In fission yeast, the heterochromatin landscape is regulated by diverse environmental signals such as nutrition and temperature [5,6,42]. However, the mechanism behind heterochromatin changes is poorly understood. In this study, we found that the activation of Pka1 by the cAMP signaling pathway, in response to high glucose levels, is required to maintain the proper levels of a JmjC domain protein Epe1.

Epe1 plays an important role in shaping the heterochromatin landscape by serving as the major "eraser" of the H3K9me mark in vivo. Loss of Epe1 causes heterochromatin expansion, ectopic heterochromatin island formation, and improved heterochromatin inheritance. In contrast, overexpression of Epe1 leads to heterochromatin defects. Therefore, Epe1 protein levels need to be tightly regulated within a narrow range. Indeed, Epe1 is a target of the Cul4-Ddb1 ubiquitin E3 ligase, which mediates its degradation by the proteasome [30]. Our results demonstrate that cAMP signaling affects the translation of the *epe1+* mRNA, independent of protein degradation. Polysome profiling indicates that the translation of *epe1+* mRNA is less efficient compared with mRNAs of other genes such as the housekeeping gene *act1+* or an inducible gene *nmt1+*, and more sensitive to disruptions of cAMP signaling. The feature of *epe1+* that is responsible for this translation control seems to be within the *epe1+* coding region, making it challenging to perform further mutational analysis. Interestingly, a recent study shows that in fission yeast, ribosomes stall on tryptophan codons upon oxidative stress [43]. It is possible that the amino acid composition of Epe1 might contribute to the lower translation efficiency.

Previous studies found that heterochromatin islands are affected in low glucose conditions, but with some discrepancies. An earlier study reports an increase of H3K9me2 in heterochromatin islands during glucose starvation while a recent study reports a decrease of H3K9me2 at heterochromatin islands [42,44]. Our ChIP-seq analysis showed that average H3K9me2 levels increase in cells grown in a low glucose medium for 6 hours compared to cells grown in a high glucose medium (Fig 6D). The differences in results could originate from different media conditions, antibodies, or the duration of exposure to low glucose. We also note that our study used glycerol as an alternative carbon source to allow cell growth in the low glucose medium. Our finding that many islands show an increase in H3K9me2 is consistent with the reduction of Epe1 levels. However, low glucose likely affects heterochromatin islands in multiple ways besides its effect on Epe1 levels as glucose starvation changes not only cAMP signaling but also the levels of many metabolites, which could affect diverse histone-modifying activities. Therefore, the effects on heterochromatin islands will reflect the sum of diverse contributions.

Recent studies have begun to tackle the mechanistic link between nutrient conditions and changes in heterochromatin. For example, the TOR signaling pathway promotes the stability of Pir1, a component of the RNA elimination machinery involved in facultative

heterochromatin formation [42]. Our data reveal a new link between glucose sensing, cAMP signaling, and the protein levels of a JmjC protein Epe1. This fits into a growing body of evidence that nutritional conditions not only change cellular metabolite levels but also affect signaling pathways to modify chromatin. Many of these regulatory events are at the post-transcriptional level, which allows cells to quickly respond to stimuli.

## Materials and methods

### Fission yeast strains and genetic analyses

Yeast strain containing *3HA-epe1*+ was constructed using the *SpEDIT* CRISPR method [45]. Guide RNA was designed using CRISPR4P [46]. Deletion strains *git1Δ*, *git3Δ*, *git5Δ*, *gpa2Δ*, *pka1Δ*, and *cgs2Δ* were derived from the Bioneer deletion library and the absence of the gene coding regions was confirmed by PCR analyses. The strains containing *cgs1Δ* or *cyr1Δ* were constructed by a PCR-based module method [38]. All other strains were constructed by genetic crosses. A list of yeast strains used is provided in S1 Table. Yeasts were grown in EMM (Edinburgh minimal medium, MPBio, 4110022,) or YEA (Yeast extract with adenine, 0.5% yeast extract, 3% glucose, and 100 mg/l adenine). For glucose deprivation experiments, yeast cells were grown first in YEA medium, washed twice with water, and resuspended in a low glucose medium (YEA with 0.1% glucose and 3% glycerol) and grown for 6 hours before ChIP analyses or RNA extraction. For serial dilution plating assays, ten-fold dilutions of a mid-log phase culture were plated on the indicated media and grown for 4–6 days at 30˚C for EMM-based plates and 3 days at 30˚C for other plates.

### Screen for suppressors of Epe1 overexpression

Query strain (*nmt41-epe1*+*-natMX6 otr:ura4*+*-hphMX6*) was crossed with a library of strains that contains individual gene deletions marked with *kanMX6* cassette, using a Singer RoToR HDA pinning robot. The desired haploid progenies, which contain *nmt41-epe1*+*-natMX6 otr*:*ura4*+*-hphMX6* and a single gene deletion were selected and pinned first onto EMM plates for 1 day to induce *nmt41* expression, and subsequently pinned onto EMM plates supplemented with 100 μg/ml FOA to measure growth.

### Chromatin immunoprecipitation (ChIP) analyses

ChIP experiments were performed as described previously [47]. Briefly, log-phase yeast cells were crosslinked with 3% formaldehyde for 30 minutes at 18˚C with shaking. Cells were harvested, washed with PBS (phosphate-buffered saline), and resuspended in ChIP lysis buffer (50 mM HEPES-KOH, pH 7.5, 140 mM NaCl, 1% Triton X-100, 0.1% Deoxycholate, 1mM PMSF). Cells were disrupted with glass beads in a bead beater. The lysates were collected, diluted with ChIP lysis buffer, and sonicated with a Bioruptor Pico (Diagenode) for 10 cycles (30s on/30s off) to produce DNA fragments of 100–500 bp in length. The cleared lysates were incubated with the following antibodies overnight at 4˚C: H3K9me2 (Abcam, 115159), H3K14ac (Upstate, 07–353), and Pol II S2P (Abcam, ab5059). Dynabeads™ Protein G (Thermo Fisher Scientific, 10004D) was then added to capture the antibodies and associated chromatin fragments. The beads were washed with ChIP lysis buffer twice, ChIP lysis buffer containing 0.5 M NaCl, Wash buffer (10 mM Tris, pH 8.0, 250 mM LiCl, 0.5% NP-40, 0.5% Deoxycholate, 1 mM EDTA), and TE (50 mM Tris pH 8.0, 1 mM EDTA). The bound chromatin fragments were eluted with TES (50 mM Tris pH 8.0, 1 mM EDTA, 1% SDS) at 65˚C for 10 minutes with shaking. The crosslinking was reversed by incubating at 65˚C overnight. The protein-DNA mixture was treated with proteinase K, and DNA was purified by phenol:

chloroform extraction and ethanol precipitation. Quantitative real-time PCR (qPCR) was performed with Luna Universal qPCR Master Mix (NEB, M3003X) in a StepOne Plus Real-Time PCR System (Applied Biosystems). DNA serial dilutions were used as templates to generate a standard curve of amplification for each pair of primers, and the relative concentration of the target sequence was calculated accordingly. A $leu1^+$ or $act1^+$ fragment was used as a reference to calculate the enrichment of ChIP over WCE for H3K9me2. A list of DNA oligos used is provided in S2 Table.

### ChIP-seq

Log-phase yeast cells were crosslinked with 1% formaldehyde for 20 minutes with shaking at room temperature, followed by 5 minutes quenching with 125mM glycine. Cells were harvested, washed with PBS (phosphate-buffered saline), and resuspended in ChIP lysis buffer (50 mM HEPES-KOH, pH 7.5, 140 mM NaCl, 1% Triton X-100, 0.1% Deoxycholate, 1mM PMSF). Ice-cold glass beads were added, and the mixtures were vigorously disrupted in a bead-beater with four 30 s rounds. The lysates were collected, to which NP buffer was added (10 mM Tris, pH 7.4, 1 M sorbitol, 50 mM NaCl, 5 mM $MgCl_2$, 1 mM $CaCl_2$). MNase was added to the reaction and the reactions were incubated at 37˚C for 20 minutes. MNase amount was titrated empirically so that the chromatin was digested to yield mainly mono- and di-nucleosomes. The reaction was stopped by adding 10 mM EDTA, and the tubes were placed on ice. 5X ChIP lysis buffer was added to the reaction, mixed by short vertexing, and the tubes were incubated on ice for 30 minutes. The reactions were then cleared by centrifugation at 16,000 x $g$ for 10 minutes. 4% of the cleared supernatant was reserved as input and the rest was used for immunoprecipitation. The protocols for immunoprecipitation, reverse-crosslinking, and DNA precipitation were as in the previous ChIP section. The precipitated DNA was treated with RNase A (Thermo Fisher Scientific, EN0531) for 1 hour at 37˚C. DNA concentration was determined with the Qubit dsDNA HS Assay Kit (Thermo Fisher Scientific, Q33230). 5–10 ng of ChIP and input DNA were used for library construction using the NEBNext Ultra II DNA Library Prep Kit for Illumina (NEB, E7645). Libraries were pooled and sequenced on a NextSeq500/550 with the Mid-output kit (150 cycles, single-end) at the JP Sulzberger Genome Center at Columbia University.

Sequencing reads were de-multiplexed and aligned to the *S. pombe* reference genome (ASM294v2), obtained from Pombase [48] with Bowtie2 using default parameters [49]. Peaks were called with MACS2 [50], and only peaks appearing in both replicates were included for downstream analysis. Genome-wide coverage was calculated with deepTools2 [51] and normalized to counts per million (CPM). The two replicates were merged to yield the average track. The coverage plot was visualized with IGV [52]. The heatmap and average profile plot were generated by deepTools2 using the union of peaks present in all strains/conditions. ChIP-seq experiments were performed in duplicates for each genotype.

### RNA analyses

RNA analyses were performed as described [28]. Briefly, RNA was extracted from log-growth phase yeast cultures using MasterPure Yeast RNA Purification Kit (Epicentre). RT-qPCR analyses were performed with Luna Universal One-Step RT-qPCR Kit (NEB, E3005L) in a StepOne Plus Real-Time PCR System (Applied Biosystems). RNA serial dilutions were used as a template to generate the standard curve of amplification for each pair of primers, and the relative concentration of the target sequence was calculated accordingly. An $act1^+$ fragment served as a reference to normalize the concentration of samples. The concentration of each target in wild type was arbitrarily set to 1 and served as a reference for other samples.

## Protein extraction and western blot analysis

Protein extraction was performed either using the bead-beating method or a NaOH-TCA method [53]. For the bead-beating method, log-phase yeast cells were harvested and lysed by beads-beating following the same lysis protocol as in ChIP. The resulting ~100 μl lysate was diluted with 300 μl ChIP lysis buffer and mixed by vertexing. An aliquot of the lysate was mixed with an equal amount of 2X SDS loading buffer and boiled for 10 minutes at 80°C. The boiled lysate was centrifuged at 16,000 x $g$ and 8 μl supernatant was separated on an SDS-PAGE gel, followed by the transfer of the proteins to a PVDF membrane. The membrane was blotted with antibodies against Tubulin (Gift from Keith Gull) [54] and HA (Santa Cruz, sc7392). The membrane was visualized using a ChemiDoc imaging system (BioRad). For the cycloheximide-chase experiment, cycloheximide was added to the medium to a final concentration of 0.15mg/ml. $2x10^7$ cells were harvested at the indicated time points and protein was extracted with the NaOH-TCA method. Western blot quantification was performed with ImageJ.

## Polysome profiling

Approximately $5 \times 10^8$ of yeast cells were lysed using a Fast Prep machine in polysome lysis buffer (20 mM Tris-HCl pH 7.5, 10 mM magnesium chloride, 50 mM potassium chloride, 10 μg/ml cycloheximide, 1 mM PMSF, 1x Halt protease and phosphatase inhibitor cocktail (ThermoFisher Scientific, 78442)). Lysate was cleared by centrifugation at 4°C at 20,000g for 10 minutes. Lysate was loaded on a 10% to 50% sucrose gradient in polysome lysis buffer. Gradients were centrifuged for 2 hours at 37,000 rpm in a Beckman SW41Ti rotor. Fractions were collected with a BioComp gradient station and a BioComp TRiAX flow cell monitoring continuous absorbance at 260 nm. To each fraction, an equal volume of phenol:chloroform pH 4.5 was added and fractions were flash frozen. For RNA extractions, the fractions were placed in a 65°C water bath and vortexed frequently for 30 minutes. The fractions were then extracted twice with phenol-chloroform and once with chloroform, and RNA was extracted with isopropanol precipitation.

## In vitro phosphorylation assay

In vitro phosphorylation was carried out in phosphorylation assay buffer (25mM pH7.5 Tris-HCl, 10mM $MgCl_2$, 1mM Dithiothreitol, 100μM ATP and 5μCi γ-32P-ATP) with recombinant Epe1, Pka1 and Rst2 fragment. Reactions were carried out at 30°C for 30 minutes with mild shaking. To the reaction, 5μl 5X SDS loading buffer was added and incubated at 80°C for 10 minutes to stop the reaction. Reactions were separated on an SDS-PAGE gel. The gel was dried in a gel dryer at 80°C for 1 hour. The dried gel was exposed to a Phosphor Storage Screen (GE) and the screen was imaged using a Typhoon Imager (GE).

## FACS analysis

Cells containing TetR-Clr4-SET and tetO-ura4-GFP reporter were cultured and kept in logarithm phase, and were harvested at various time points after the addition of tetracycline (2.5mg/ml). Cells were collected and fixed by the addition of 70% ethanol for 20 minutes. The cells were then washed twice with PBS (10 mM $Na_2HPO_4$, 1.8mM $KH_2PO_4$, pH 7.4, 137 mM NaCl, 2.7mM KCl), and resuspended in a FACS tube (BD Falcon). GFP fluorescence was measured using FACSCelesta (Becton Dickinson), and excitation was achieved by using an argon laser emission of 488 nm. Data collection was performed using Cellquest (Becton Dickinson), and a primary gate based on physical parameters (forward and side light scatter) was set to

exclude dead cells or debris. Typically, 50,000 cells were analyzed for each sample and time point. Raw data were processed and histograms were drawn using FlowJo (10.6.2, Becton Dickinson).

## Supporting information

**S1 Fig. Pka1 phosphorylates Epe1.** (A) *In vitro* kinase assay measuring Pka1's activity towards Epe1 and a positive control Rst2 (1–380). Left, recombinant full length Epe1 purified from insect cells were used. Note that Epe1 is phosphorylated without adding recombinant Pka1, suggesting that insect cell lysates contain a kinase activity that phosphorylates Epe1. Right, recombinant Epe1 fragments purified from E.coli were used. * represents Pka1 autophosphorylation and arrows represent Epe1 fragments. (B) Diagram of Epe1 truncations and their phosphorylation status in (A). (C, D) Western blots to measure Epe1 mutants levels and tubulin. S717 is a phosphorylation site identified by mass-spec analysis of in vitro phosphorylated recombinant Epe1. S606 and T607 are predicted Pka1 phosphorylation sites based on bioinformatics analysis.
(EPS)

**S2 Fig. Endogenous HA tagged Epe1 is functional.** (A) Diagram of the mating-type region with *SacI::ade6+* reporter gene. The shaded area represents heterochromatin. (B) Ten-fold serial dilution analyses to measure the expression of *SacI::ade6+*.
(EPS)

**S3 Fig. Defective cAMP signalling has no impact on the translation of Clr4.** (A) qRT-PCR analysis of *clr4+* transcripts from each fraction of polysome profile. Relative amounts of transcripts were calculated using the delta Ct method. The distribution is shown as a percentage of the total. Error bars represent the SD of two technical replicates. (B) Western blot analysis to measure the levels of endogenous Myc-tagged Clr4 and tubulin.
(EPS)

**S4 Fig. The effects of cAMP signaling and extracellular glucose concentration on the inheritance of an ectopic heterochromatin.** (A) Diagram of the TetR-Clr4-SET system. The targeting of the SET domain of Clr4 to *tetO* sites results in the formation of heterochromatin and silencing of the adjacent *gfp+* reporter gene. The addition of tetracycline (TET) results in the release of Clr4 from *tetO* sites and heterochromatin is maintained by endogenous Clr4, which contains a chromodomain that recognizes H3K9me. (B,C) FACS analyses of GFP expression. Samples were taken at indicated time points after the addition of tetracycline.
(EPS)

**S5 Fig. Extracellular glucose concentration affects Epe1 protein levels.** (A) Western blot analyses to measure the levels of 3HA-tagged Epe1 and tubulin. (B) Quantification of Epe1 proteins levels. Error bars represent SD, n = 3.
(EPS)

**S1 Table. Yeast strains used in this study.**
(XLSX)

**S2 Table. DNA oligos used in this study.**
(XLSX)

## Acknowledgments

We thank Karl Ekwall for strains, Rodney Rothstein, Laura Landweber, Marko Jovanovic, and members of the Jia lab for discussions, and Qiulin Zhu for editing the manuscript.

## Author Contributions

**Conceptualization:** Kehan Bao, Chun-Min Shan, Songtao Jia.

**Data curation:** Kehan Bao, Chun-Min Shan, Xiao Chen, Gulzhan Raiymbek, Jeremy G. Monroe, Yimeng Fang, Takenori Toda, Kristin S. Koutmou, Kaushik Ragunathan, Luke E. Berchowitz, Songtao Jia.

**Funding acquisition:** Kristin S. Koutmou, Kaushik Ragunathan, Chao Lu, Luke E. Berchowitz, Songtao Jia.

**Investigation:** Kehan Bao, Chun-Min Shan, Xiao Chen, Gulzhan Raiymbek, Jeremy G. Monroe, Yimeng Fang, Takenori Toda, Luke E. Berchowitz.

**Methodology:** Kehan Bao, Chun-Min Shan.

**Supervision:** Kristin S. Koutmou, Kaushik Ragunathan, Chao Lu, Luke E. Berchowitz, Songtao Jia.

**Writing – original draft:** Kehan Bao.

**Writing – review & editing:** Kehan Bao, Chun-Min Shan, Yimeng Fang, Takenori Toda, Kaushik Ragunathan, Chao Lu, Luke E. Berchowitz, Songtao Jia.

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
