## [Decision Letter · Decision Letter 0]

25 Oct 2021

Dear Dr Jia,

Thank you very much for submitting your Research Article entitled 'The cAMP signaling pathway regulates Epe1 protein levels and heterochromatin assembly' to PLOS Genetics.

The manuscript was fully evaluated at the editorial level and by three independent peer reviewers, all experts in the field. The reviewers appreciated the attention to an important topic but identified concerns that we ask you address in a revised manuscript.

Linking a pathway sensitive to nutritional state to the formation of heterochromatin is an important contribution, and therefore all three reviewers are overwhelmingly positive in their remarks. They deem your work carefully planned and conducted, and concluded that the data provided support your conclusions. Relatively minor changes will need to be made before this manuscript may be acceptable for publication at PLOS Genetics. Addressing comment 1 of Reviewer 2 may require additional experimentation. Reviewer 3 had comments addressing results that were obtained based on the Materials and Methods but were not shown or discussed in the manuscript (comment regarding ChIP-seq data). Similarly, data that are not shown and not provided as supplement are briefly discussed; this practice (i.e., "data not shown") is generally discouraged and we suggest that you either provide the data (even though they may be "inconclusive") or re-phrase the sections mentioned by Reviewer 3. All other reviewer comments are relatively minor and should be addressed in your point-for-point response letter.

[LINK]

Yours sincerely,

Michael Freitag

Associate Editor

PLOS Genetics

John Greally

Section Editor: Epigenetics

PLOS Genetics

Reviewer's Responses to Questions

**Comments to the Authors:**

Reviewer #1: The paper by Jia and colleagues identifies the cAMP-PKA signaling pathway as a regulator of translation efficiency of Epe1, a putative H3K9me demethylase that negatively regulates heterochromatin spreading. Thus, the authors identify a link between a nutritional-sensitive pathway and he formation of heterochromatin.

The authors employ the S. pombe deletion library to isolate mutant strains that affect heterochromatin at the centromere. The query strain contains a mild overexpression of Epe1 (nmt41-Epe1) and a reporter gene at the centromeric heterochromatin (otr4::ura4+). They identify 5 deletion mutations in the cAMP/PKA signaling pathway. Analysis of the transcription levels of pericentric repeats (dh) suggests that loss of cAMP/PKA signaling affects heterochromatin formation in cells overexpressing Epe1 at a natural chromosomal context (Fig. 1-2E). The authors go on to show a correlation between intact cAMP/PKA signaling and the protein level of Epe1 (Fig. 2F). Using polysome profiling, the authors demonstrate a decrease in epe1+ mRNA in the polysome fractions in git3 mutant cells, suggesting that cAMP/PKA affects the translation efficiency of epe1+ mRNA.

Furthermore, the authors show that the cAMP-PKA pathway affects the endogenous levels of Epe1 (Fig. 4A) and that git3 mutant cells, like epe1 mutant cells, affect heterochromatin spreading outside the mating type locus and increase H3K9me2 at heterochromatic islands. Consistent with the idea that cAMP/PKA affects Epe1 levels, the authors show that in low glucose, the protein levels of Epe1 are reduced and there is a slight increase in HE3K9me at heterochromatin islands.

The data is generally of high quality and the findings are novel and of interest to the field of chromatin biology. To my opinion, the main weakness of the article is the limited analysis of the effect of the cAMP-PKA signaling on wild type cells expressing endogenous levels of Epe1 (see comment 3 and 4 below).

Concerns:

1. Fig. 2E. The authors suggest that "cgs2Δ, cgs1Δ, or the addition of cAMP have little effects on dh transcript". Statistical analysis is missing.

2. Fig. 3C. It is not clear to me how the experiment was performed, since the level of Epe1 at time 0 is similar in WT and git3 cells (but Epe1 levels are reduced in git3 cells, as shown in Fig. 3B).

3. What is the effect of deletion of git3 on the steady state levels of endogenous Epe1? And/or:

4. What is the effect of deletion of git3 on translation efficiency of epe1+ mRNA in wild type cells?

5. There is relatively little data to show that low glucose affects heterochromatin formation (Fig. 4 E, G). The authors may consider adding the effect of mutating cgs2 on the levels of Epe1 in low glucose (Fig. 4E). In addition, does low glucose affect heterochromatin spreading outside the mating type locus (similar to the effect of the epe1 mutation, Fig. 4C)?

6. The manuscript should be carefully edited for typos.

Reviewer #2: Bao et al. use a genetic screen to search for suppressors of Epe1 overexpression-mediated loss of heterochromatin and gene silencing. Epe1 is a JmjC domain anti-silencing factor that promotes histone H3K9 demethylation, and plays a major role in preventing inappropriate H3K9me3 spreading and epigenetic maintenance. The authors cross nmt41-epe1 overexpressing cells to the S. pombe deletion library for non-essential genes to search for genes that may regulate Epe1 expression or function. Their screen uncovers multiple genes in the cAMP signaling pathway and they go on to demonstrate that cAMP signaling is required for high level expression of both nmt41-driven and endogenous Epe1. They further show that cAMP signaling acts through the Pka1 kinase to promote efficient translation of epe1+ mRNA. They conclude that the cAMP pathway regulates heterochromatin formation in response to changes in nutritional signals. The experiments are straightforward, well-executed, and support the main conclusion of the paper. Altogether, the study provides new insight into how changes in signaling pathways linked to growth conditions and metabolism regulate the expression of the JmjC domain protein Epe1 and provides evidence that this regulation takes place at the level of translation. The authors should consider the following points.

1. Is the regulation of translation by cAMP signaling general or specific to epe1+. The only controls the authors provide, nmt1+ and act1+ polysome associations (Figure 3D), seem to indicate that in git3-delta cells, polysome association of both these control mRNAs is reduced. Both nmt1+ and act1+ are highly abundant mRNAs. Have the authors tested the effect of deleting git3 on a few other mRNAs that are lowly expressed? This would be a valuable addition to the data in Figure 3.

2. Epe1 has also been shown to play a critical role in DNA sequence-independent epigenetic inheritance of H3K9me and heterochromatin (PMID: 25831549, PMID: 25838386). It would be great to know whether attenuation of cAMP signaling would allow epigenetic inheritance of heterochromatin in epe1+ cells.

3. The authors present ChIP-seq data showing increased H3K9me2 levels at the mating type locus when cells are grown in low glucose. Does LG affect H3K9me2 spreading at other loci or the silencing of the Sac1::ade6+ reporter? Does low LG allow sequence-independent heterochromatin inheritance?

Was the ChIP-seq in Fig. 4G performed on multiple clones to ensure that the authors are not looking at small sample to sample variations in H3K9me2 levels?

Minor

-Does the 3X HA tag affect Epe1 function?

Reviewer #3: General comments:

Environmental changes lead to transcriptional changes. However, the role of chromatin in this process is not well understood. In this study, the authors devise a clever genetic screen to explore how the chromatin modifier, Epe1, regulates a transcriptionally repressive chromatin mark (H3K9me2) in Schizosaccharomyces pombe. Surprisingly, the authors find components of the conserved cAMP signaling pathway regulate the activity of Epe1. Using the powerful genetics of S. pombe, the authors demonstrate that cAMP signaling has an important role in the efficient translation of epe1 mRNA, maintenance of Epe1 levels, and regulation of chromatin landscape in response to environmental changes. In summary, the authors present a very interesting study that demonstrates a novel link between environmental signaling and chromatin regulation.

The manuscript is within the scope of PLOS Genetics.

The authors should add line and page numbers to aid the review process.

The authors should check comma usage throughout the manuscript and ensure manufacturer information is provided and consistent for all commercial products.

Major comments:

The third paragraph in the discussion walks through experiments that were conducted during the study. These experiments are very interesting and a logical next step for this work. However, the authors discuss data not included in the manuscript due, at least partially, to lack of robustness (mass spec data, i.e., “We could not rule out the presence of additional sites because the peptide coverage was not 100% and the phosphorylation is weak.”). Results should not be presented in the discussion particularly in the absence of data. If results are robust enough to discuss, they (and the underlying data) should be presented in the results section.

In materials and methods, the authors state how peaks were called (MACS2) and heatmaps were generated (deeptools2); however, it is unclear that this data is presented in this manuscript. It would be interesting to see additional results from the ChIP-seq data, i.e., how individual heterochromatin islands change or if new H3K9me2 peaks were observed. If these results were assessed, are robust, and provide additional evidence in support of this study, they should be presented in the results section. Otherwise, materials and methods should be changed accordingly.

Additional comments:

Abstract:

The following sentence overstates the impact of this study and should be rewritten: “These results provide a mechanism for the regulation of heterochromatin in response to changes in nutritional conditions.”

Results:

The following sentence should be rewritten for clarity: “In the presence of cAMP, Cgs1 dissociates from Pka1, leading to Pka1 translocation to the nucleus and its activation to phosphorylates its substrates.”

Based on the data presented and inherent limitations with ChIP-seq, it is difficult to claim that “average H3K9me2 levels increase” in low glucose conditions (a quantitative ChIP-seq approach could provide support for the claim). A claim that “average H3K9me2 levels appear to increase” is reasonable. Additionally, presenting data for individual heterochromatin islands could provide clearer results. Of particular interest would be ChIP-seq data for the ssm4 and mei4 loci, which are quite different in relevant mutants in ChIP-qPCR experiments. See the following sentence: “ChIP-seq analyses show that heterochromatin islands positions are maintained and average H3K9me2 levels increase in cells grown in a low glucose medium for 6 hours compared to cells grown in the rich medium (Fig. 4G), consistent with the reduction of Epe1 levels.”

Materials and Methods:

Media (plural) should be replaced with medium (singular) in the following sentence and throughout the manuscript as needed: “Yeasts were grown in EMM media or YEA media.” Additionally, the authors should define abbreviated media and provide recipes or references for media used in this study.

The authors should provide additional information on the source of the tubulin antibody used in this study. As described, these experiments could not be replicated. See the following sentence: “The membrane was blotted with antibodies against Tubulin (gift) and HA (sc7392, Santa Cruz).”

The authors should clarify the meaning of “25 OD” in the following sentence: “25 OD of yeast cells were lysed using a Fast Prep machine in polysome lysis buffer (20 mM Tris- HCl pH 7.5, 10 mM magnesium chloride, 50 mM potassium chloride, 10 μg/ml cycloheximide, 1 mM PMSF, 1x HaltTM protease and phosphatase inhibitor cocktail (ThermoFisher Scientific, 78442).” Additionally, the first parenthesis is not closed.

Figures and Tables:

Figure 1C: Some protein names in the figure are in bold. These appear to be components that were identified in the genetic screen. The authors should describe what is represented by proteins in bold in the figure legend.

Figure 4D: The authors state that three loci were tested. However, results for only two loci are presented (and mentioned in results). See the following sentence: “ChIP analyses of H3K9me2 levels at three heterochromatin island loci.”

Figure 4G: The authors should provide additional information pertaining to what is shown in panel G.

**Have all data underlying the figures and results presented in the manuscript been provided?**

Reviewer #1: Yes

Reviewer #2: Yes

Reviewer #3: Yes

PLOS authors have the option to publish the peer review history of their article (what does this mean?). If published, this will include your full peer review and any attached files.

Reviewer #1: No

Reviewer #2: No

Reviewer #3: No

---

## [Editor Report · Decision Letter 1]

20 Jan 2022

Dear Dr Jia,

We are pleased to inform you that your manuscript entitled "The cAMP signaling pathway regulates Epe1 protein levels and heterochromatin assembly" has been editorially accepted for publication in PLOS Genetics. Congratulations!

In your responses to reviewers and in the revised version of the manuscript you more than adequately addressed all reviewer comments. Adding additional experimental evidence makes this a much stronger contribution!

Yours sincerely,

Michael Freitag

Associate Editor

PLOS Genetics

John Greally

Section Editor: Epigenetics

PLOS Genetics

Comments from the reviewers (if applicable):

**Data Deposition**

http://datadryad.org/submit?journalID=pgenetics&manu=PGENETICS-D-21-01156R1

**Press Queries**

---

## [Editor Report · Acceptance letter]

9 Feb 2022

PGENETICS-D-21-01156R1 

The cAMP signaling pathway regulates Epe1 protein levels and heterochromatin assembly 

Dear Dr Jia, 

We are pleased to inform you that your manuscript entitled "The cAMP signaling pathway regulates Epe1 protein levels and heterochromatin assembly" has been formally accepted for publication in PLOS Genetics! Your manuscript is now with our production department and you will be notified of the publication date in due course.

With kind regards,

Katalin Szabo

PLOS Genetics

On behalf of:
